# Effect of Thermal Treatment on the Internal Structure, Physicochemical Properties and Storage Stability of Whole Grain Highland Barley Flour

**DOI:** 10.3390/foods11142021

**Published:** 2022-07-08

**Authors:** Bin Dang, Wen-Gang Zhang, Jie Zhang, Xi-Juan Yang, Huai-De Xu

**Affiliations:** 1College of Food Science and Engineering, Northwest A & F University, Xianyang 712100, China; 2008990019@qhu.edu.cn; 2Qinghai Tibetan Plateau Key Laboratory of Agricultural Product Processing, Academy of Agriculture and Forestry Sciences, Xining 810016, China; 2017990098@qhu.edu.cn (W.-G.Z.); 2015990070@qhu.edu.cn (J.Z.); 3Academy of Agriculture and Forestry Sciences, Qinghai University, Xining 810016, China

**Keywords:** highland barley, microstructure, gelatinization, functional properties, storage stability

## Abstract

In this study, to improve the processing performance of whole grain highland barley flour (whole grain HB flour), they were prepared by sand-roasting, far-infrared baking, steam explosion, and extrusion, and the effects of on functional properties and storage characteristics were measured. The results indicated that sand-roasting, far-infrared baking, and steam explosion all caused cracks and honeycomb structures in the outer layer and endosperm of the highland barley. The XRD analysis results indicated that highland barley starch treated by far-infrared baking exhibited typical A-type crystal structure, while sand-roasting, steam explosion, and extrusion presented the typical V-type. The results of DSC analysis revealed that the onset temperature (To), peak temperature (Tp), gelatinization enthalpy (ΔH), peak viscosity (PV), trough viscosity (TV), and final viscosity (FV) decreased significantly, while the swelling power, water-holding capacity and oil-holding capacity increased significantly. During the storage period, the moisture content and lipase activity of the whole grain HB flour after thermal treatment remained at a low level; the fatty acid value, peroxide value, and malondialdehyde value increased; finally, the cooked whole grain HB flour was unstable during storage. The functional properties of whole grain HB flour can be improved by steam explosion, and will then have better storage stability.

## 1. Introduction

Highland barley is the most distinctive food crop on the Qinghai Tibet Plateau, and it is also the primary food crop for the survival of farmers and herdsmen in Tibet. Highland barley has high protein, high dietary fiber, high vitamins, low fat and low carbohydrates [1]. Its aleurone layer and germ are rich in linoleic acid, dietary fiber (DF), resistant starch (RS), vitamins, gamma-aminobutyric acid (GABA), β-glucan, phenolic compounds, and other nutritional chemical components, and such components endow highland barley with the functions of antioxidation, human glucose and lipid metabolism regulation, antitumor, anti-cardiovascular disease and immunity improvement [1]. Therefore, the development of whole grain highland barley products is the primary method of ensuring the health role of highland barley. However, due to its hard texture, thick seed coat, difficult gelatinization, rough taste, and poor processing performance, the application of highland barley for food processing in the form of whole grain is greatly limited [1]. For these reasons, the development of ready-to-eat food by improving the quality and processing of whole grain highland barley products with modern processing technology has become a research hotspot of whole grain food.

It is an economical and feasible product preparation method to cook highland barley grains by thermal treatment, then grind them [2]. Thermal treatment realizes better texture, crispier taste, and higher volume of whole grain. It also prolongs shelf life, enhances flavor and improves human digestibility [3]. For these reasons, cooked flour has become the main ingredient for developing instant and convenient whole grain products. In general, the main methods of cooking grain are baking, sand-roasting, steam explosion, extrusion, and microwave heating [4,5]. Among these, sand-roasting is a traditional method of grain cooking. The grain is stir-fried at high temperature under the hot air condition of normal atmospheric pressure, and the water vaporizes in situ, resulting in the puffing of grain [6]. The Tibetan people of China stir-fry highland barley grains to make flour, and then add an appropriate amount of yak butter, milk residue, and sugar or salt to make Zanba, a type of traditional food that is very popular in Tibet. It is a type of instant food that has unique local characteristics [1]. Studies have indicated that this method can improve the color, aroma, taste, and texture of agricultural products, increase their volume, crispness, digestibility, and shelf life [5,6], as well as improve the chewiness of highland barley [7], the content of free phenol, and the extraction rate of β-glucan [8]. With the advantages of short treatment time and high efficiency, infrared heating will cause changes in the porous structure and physical properties such as swelling, bulk density, hardness, and color of shell-less barley [9]. Due to the sudden release of high temperature and high pressure in explosion puffing, the cell wall of grain is broken to form a loose and porous structure, and the nutrients in the cells are released [10]. Consequently, explosion puffing will reduce the content of ochratoxin A in rice and oats [11], promote the release of phenolic substances [12], β-glucan, pentosan, and vitamin E in highland barley, and improve its antioxidant activity [13]. Extrusion is a continuous process making use of high temperature, high pressure, and high shear conditions [14], and will overcome the shortcomings of rough taste and difficult digestion of grains, alter the nutritional components of starch and protein in grains, and be more conducive to human digestion and absorption [15]. All the above methods can make the grain expand, cause starch gelatinization, alter the morphology and structure of grain, improve the taste, increase the digestibility, reduce harmful substances, and promote the release of functional components [2,8]. However, most of them focus on the tissue structure and physical properties of cooked whole grain, while there have been few studies performed on the processing characteristics of cooked whole grain.

The application of cooked whole grain flour in food primarily depends on the thermal treatment method and its influence on the processing and storage characteristics of whole grain flour. A limited number of studies have reported that baking reduces the bulk density, emulsifying ability, and foaming ability of baked wheat flour, alters the gelatinization characteristics of flour, and reduces the peak value, valley value, breakdown viscosity, and final viscosity [16]. Baked barley flour has a significant water absorption and water solubility index. Baking significantly affects the gelatinization and thermal properties of flour and increases the content of cooked starch [17]. Steam explosion will lead to significant changes in the structure and physical properties of grain starch and improve water absorption and water-holding capacities of cereal grain and flour [8]. Wang Lili compared and concluded that microwaving, baking, and superheated steam treatment reduced the breakdown viscosity of whole grain HB flour, while also killing enzymes in highland barley grain, resulting in changes in its structure, physicochemical properties, and functional properties [7]. In addition, different thermal treatments can change the structure of highland barley starch and protein, and alter the physicochemical properties of highland barley flour [2]. baking treatment increased the gelatinization temperature of highland barley starch and reduced its swelling power, solubility, gelatinization enthalpy, and gelatinization viscosity [4]. It is worth noting that most existing studies on the processing characteristics of grain flour by thermal treatment have been performed with a single processing technology, while there are few reports with respect to the comparison of sand-roasting, far-infrared baking, steam explosion, and extrusion. Presently, the effects of thermal treatment on the structure and physicochemical properties of highland barley flour have been studied from the perspective of the enzyme-destroying effect, and the flour is not fully cooked, therefore all of them focused on the effect of thermal treatment on the structure of highland barley starch and protein. However, research on the functional properties and storage characteristics of cooked highland barley flour after thermal treatment has not yet been reported. Only by clarifying the physicochemical and functional properties and the storage characteristics of cooked highland barley flour can appropriate treatment methods be selected according to the processing requirements of different highland barley products.

Therefore, using untreated highland barley flour as the control, this paper studies the effects and differences of four thermal treatment technologies of sand-roasting, far-infrared baking, steam explosion, and extrusion on the structure, physicochemical, functional properties, and storage characteristics of whole grain HB flour, so as to provide a theoretical basis for the quality improvement of whole grain HB flour and its application in food development.

## 2. Materials and Methods

### 2.1. Highland Barley

Kunlun 15 (white-grain) highland barley is grown in Xining City. The highland barley seeds harvested in August were provided by the Qinghai Academy of Agriculture and Forestry Sciences, and the basic nutrients are as follows: protein is 10.56%, total starch is 49.58%, crude fiber is 2.44%, crude fat is 2.06%, ash is 1.59%, β-glucan is 6.07%), stored at −18 °C. All other chemical reagents were of analytical grade.

### 2.2. Sample Preparation

Each method was pretested to determine the conditions of sand-roasting, far-infrared baking, steam explosion and extrusion. In each case, the grain was completely cooked. The specific treatment is described as follows:

Sand-roasting: Soak the highland barley grains in water for 6 h until the moisture content reaches 40%, drain them and place them in a constant-temperature automatic stir-frying machine (CH50, Henan Huian Mechanical Equipment Co., Ltd., Zhengzhou, Chnia), stir-fry them at 110 ± 5 °C for 10 min, cool the fried highland barley grains to room temperature (20–25 °C), and place them in a polyethylene bag for standby.

Far-infrared baking: Soak the highland barley grains for 6 h until the moisture content reaches 40%, spread them flat on the oven tray in a thin layer (thickness of 0.5 cm), and bake them with a CK-2 far-infrared food oven (Guangzhou Maisheng Baking Equipment Co., Ltd., Guangzhou, China) at a bottom heating temperature of 150 ± 5 °C and upper heating temperature of 170 ± 5 °C for 20 min, then cool them to room temperature (20–25 °C) and place them in a polyethylene bag for standby.

Steam explosion: Adjust the moisture content of highland barley to 10% before steam-explosion treatment. Place 1 kg highland barley grains in an air puffer (XSS-QPD large air puffer, Wuhan Xinshishang Food Machinery Co., Ltd., Wuhan, China) with a pressure of 1.25 MPa. After 7 min, conduct air puffing. Cool the puffed highland barley grains to room temperature (20–25 °C) and place them in a polyethylene bag for standby.

Extrusion: After grinding the highland barley grains, add 34% pure water and process the highland barley grains into highland barley chips using a twin-screw extruder (DZ65-Ⅱ, Jinan Saixin Machinery Co., Ltd., Jinan, China). The extrusion temperature should be 55 °C in zone I, 180 °C in zone II, and 160 °C in zone III; the feeding frequency should be 22 Hz, the rotary cutting frequency should be 26 Hz, and the screw speed should be 800 r/min. Cool the extruded highland barley chips to room temperature (20–25 °C) and place them in a polyethylene bag for standby.

Grind the untreated and heat-treated barley with a high-speed grinder (HK-04A, Guangzhou Xulang Mechanical Equipment Co., Ltd., Guangzhou, China) into whole grain HB flour, then determine the quality of the whole grain HB flour.

### 2.3. Whole Powder Characteristics

#### 2.3.1. Color Values

A WSC-S automatic color difference meter was used for detection. The same sample was measured 3 times, and the average value was taken to determine the color parameters of the whole grain HB flour. L* (L* = 0: black; L* = 100: white), a* (+a* = red, −a* = green) and b* (+b* = yellow, −b* = blue) were obtained. Brightness (L*), red (a*), yellow (b*), and color difference between untreated and cooked highland barley flour (ΔE) were obtained by the equation:(1)$$\Delta Ε=\sqrt{{\left(\Delta L^{*}\right)}^{2}{+(\Delta a^{*})}^{2}{+(\Delta b^{*})}^{2}}$$

#### 2.3.2. Scanning Electron Microscopy

We removed an appropriate amount of highland barley flour sample and placed it on the sample stage [18]. We sprayed gold and conducted electricity in the vacuum plating instrument. We observed the apparent structure of the sample under the working voltage of 15 kV with an electron-scanning microscope (S-3400H, HITACHI, Tokyo, Japan).

#### 2.3.3. X-ray Diffraction (XRD) Analysis

An X-ray diffractometer (X’Pert Pro, PANalytical B.V., Amsterdam, The Netherlands) was used for continuous scanning. The scanning voltage was 36 kV, the current was 20 mA, the scanning range was 5–40°, the scanning speed was 2°/min, and the increment was 0.02° (2θ). Jade 6.5 software was applied to calculate the relative crystallinity as a percentage of the diffraction peak area according to the total peak area.

#### 2.3.4. Thermal Properties

We accurately weighed a 3 mg sample and 6 mg of deionized water, placed them in an aluminum crucible, covered, and sealed them. We conducted differential scanning calorimetry measurements using a thermal analyzer (STA449F3, Netzsch Instrument Manufacturing Co., Ltd., Selb, Germany) to measure the thermal characteristics (Zhu et al., 2020). The test was performed under the condition of filling N2, with the heating rate 10 °C/min and the measurement range of 22–200 °C [13,14]. By analyzing the spectrum, the onset temperature (To), peak temperature (Tp), final temperature (Tc), and gelatinization absorption enthalpy (ΔH) were obtained.

### 2.4. Functional Properties of Whole Powder

#### 2.4.1. Determination of Tap and Bulk Densities

The dynamic viscosity of highland barley flour was measured by a rapid viscosity analyzer RVA 4500 (Perten Instruments, Sweden) [19]. The moisture standard of the sample was 14% and the sample volume was 3.50 g. The speed of the agitator was kept at 960 r/min for 10 s and 160 r/min for the remainder of the time. The data were analyzed by Thermocline for Windows (TCW) software. The gelatinization characteristics of highland barley flour indicated the peak value and rising trend curve of viscosity with heating time. The maximum and minimum viscosity during heating and the viscosity at the end of the temperature program were represented by peak viscosity (PV), holding viscosity (HV), and final viscosity (FV), respectively. Additionally, the differences between the PV and HV, and FV and HV were described as breakdown viscosity (BD) and setback value (SB), respectively.

For the tap density [20], we placed the whole highland barley flour sample (10 g) into a 25 mL cylinder and tapped the cylinder 15 times on the workbench. We measured the final volume of the sample and expressed the tap density in g/mL. For bulk density, we filled the measuring cylinder (100 mL) with the sample to the 100 mL mark, obtained the weight with digital balance (AL204, METTLER TOLEDO Instrument Co., Ltd., Shanghai, China), and expressed the bulk density in g/mL. Each sample was measured in parallel three times.

#### 2.4.2. Water- and Oil-Holding Capacity

We weighed a 2.5000 g (M_0_) sample and placed it in a centrifuge tube, weighed M_1_, added 30 mL distilled water (edible oil is also acceptable), heated with boiling water, and stirred for 15 min. We cooled it to room temperature, centrifuged it at 3000 r/min for 20 min, placed the test tube upside down on the test tube rack, let it stand for 10 min, drained the water (or edible oil), and wiped the sample dry with filter paper, then accurately weighed the mass M_2_. The WHC and OHC were calculated with the equation in the formula [21].
(2)$$\mathrm{WHC}/\mathrm{OHC}\;(g/g)=\frac{M_{2}-M_{1}}{M_{0}}$$

#### 2.4.3. Determination of Swelling Power

The method described by Crosbie, G.B. (1992) was used [22]. We accurately weighed a 1.0 g sample, placed it in a glass test tube with a scale, recorded the volume V_1_, added 10 mL distilled water, stirred evenly, let it stand at room temperature for 24 h, and read the volume of the sample as V_2_.
(3)$$\mathrm{SP}\;(\%)=\frac{V_{2}-V_{1}}{V_{1}}$$

### 2.5. Storage Stability

#### 2.5.1. Storage Experiment and Sample Preparation

For the accelerated storage test, the whole grain HB flour after thermal treatment is packed at 500 g/share under the temperature of 50 ± 1 °C and humidity of 50 ± 1% conditions for accelerated storage. The moisture, lipase activity, fatty acid value, peroxide value, and malondialdehyde value were detected every 14 days. Each measurement was 60 g, 7 times (accurate to 0.01 g). 1 day under 50 °C was equivalent to 20 °C for 8 days.

#### 2.5.2. Analysis of Lipase Activity

Lipase activity was determined according to Yu’s study [23]. We weighed 2.0000 g whole grain HB flour, added 1 mL pure oil, mixed well, then added 5 mL phosphoric acid buffer (pH 7.4) and 5 mL distilled water. After sealing, we placed it in a 30 °C incubator for 24 h. We removed it, added a 50 mL mixture of ethanol and ether (4:1), shook well, and filtered. We accurately transferred 25.0 mL filtrate to a conical flask, added 3–5 drops of phenolphthalein indicator, titrated with 0.05 mol/L potassium hydroxide solution until it was reddish and did not fade for 30 s, and recorded the volume of potassium hydroxide solution consumed. We removed another 2 g sample for a blank test. Except for the fact that it did not need to be kept at 30 °C for 24 h, all other operations were the same as above.
(4)$$X\;(\%)=\frac{(V_{1}-V_{2})\times c\times 56.1}{m\times (100-M)}\times \frac{60}{25}\times 100$$

#### 2.5.3. Fatty Acid Value Detection

We mixed 2 g whole grain HB flour with 50 mL petroleum ether, shook for 10 min, and filtered and collected the filtrate. We mixed 25 mL filtrate with 75 mL 50% ethanol solution, added 5 drops of phenolphthalein, and titrated with 0.01 M NaOH until the color turned light pink. We recorded the volume of NaOH consumed V_1_ (mL). Meanwhile, we replaced the sample with petroleum ether for a blank test, and recorded the volume of NaOH consumed V_0_ (mL) [24].
(5)$$\mathrm{Ak}\;(\%)=(V_{1}-V_{2})\times c\times 56.1\times \frac{50}{25}\times \frac{100}{m\times (100-w)}\times 100$$

#### 2.5.4. Peroxide Value

We accurately weighed 0.1000 g whole grain HB flour, mixed it into 9.8 mL chloroform methanol (7:3) solution, and shook it in a vortex mixer for 4 s. We added 50 μL ammonium thiocyanate (NHSCN) solution and shook for 4 s. Then we added 50 μL Fe+ solution and shook for 4 s. We stored it at room temperature for 5 min and centrifuged it (1000× *g*, 2 min), then took the supernatant and measured the absorbance at 500 nm with a spectrophotometer. The above solution without sample was used for the blank control. The entire measurement should be completed within 10 min [25].
(6)$$\mathrm{Peroxide}\;\mathrm{value}\;(\mathrm{meq}/\mathrm{kg})=\frac{(\mathrm{As}-\mathrm{Ab})\times m}{m_{0}\times 55.84\times 2}$$

#### 2.5.5. Malondialdehyde (MDA) Content

We Weighed 2.000 g whole grain HB flour, added 10 mL 10% trichloroacetic acid, and centrifuged it for 10 min at 4000 r/min. We removed the supernatant and diluted it to 10 mL with trichloroacetic acid. We added 2 mL 0.6% sulfurized barbituric acid. After mixing, we bathed it in boiling water for 15 min and measured the absorbance at 450, 532, and 600 nm after cooling [26].
(7)$$\mathrm{MDA}\;(\mathrm{mg}\cdot {\mathrm{kg}}^{-1})=\frac{(6.45\times (A_{532}-A_{600})-0.56\times A_{450})\times 10}{M\times 12}\times 0.072$$

#### 2.5.6. Analysis of Moisture Content

The moisture content of whole grain HB flour was determined according to the Chinese standard (GB5009.3-2016). In short, the whole grain HB flour was kept at 105 °C and dried in a drying oven (DGG-9140A, Jinghong Experimental Equipment Co., Ltd., Shanghai, China) until the sample weight was stable. The weight before and after heating was measured by electronic analytical balance (FA1104, Shanghai Jingtian Electronic Instrument Factory, Shanghai, China).

### 2.6. Statistical Analysis

All the above tests were conducted in three groups (mean value ± standard deviation). The significant difference between Duncan test and Pearson correlation was calculated by IBM SPSS 22.0. All data for this work were described using Origin 18.

## 3. Results

### 3.1. Appearance and Structure of Highland Barley Grain Subsection

Figure 1 portrays the appearance of untreated highland barley (control) and highland barley after different thermal treatment (sand-roasting, far-infrared baking, steam explosion and extrusion). It can be observed from Figure 1A–E that the external morphology of highland barley was different after different thermal treatment. Cracks (crackles) appeared on the surface of highland barley after sand-roasting and steam explosion, and the grain volume increased significantly, thus indicating that these two thermal treatment methods can destroy the surface structure of highland barley. The reason for this may lie in the fact that the liquid water inside highland barley grain quickly “flashed” into steam under high temperature conditions, resulting in grain swelling [27], which was more consistent with the existing reported results that thermal treatment can alter the appearance and structure of grain [8,9]. The surface cracks of highland barley after steam explosion were more uniform, thus the puffing effect of highland barley was more significant. There was no crack on the surface of highland barley after far-infrared baking, and it had no obvious puffing effect on highland barley grain. It has been reported that the effect of far-infrared heating on grain swelling depends on the initial moisture content of the sample. When the initial moisture content is 29.5%, the cross-section swelling of naked barley heated by far-infrared is low [9]. The grain matrix would be too soft at a high moisture content, thus preventing the sudden release of steam pressure due to the lack of pressure barrier, resulting in reduced swelling [28]. In this test, the moisture content of highland barley grain reached 40% by pretreatment, then far-infrared baking was performed. Therefore, there was no obvious puffing of highland barley grain. In extrusion, the whole grain highland barley is made into flour and then extruded by a twin-screw extruder. Therefore, the grain structure of highland barley cannot be retained. In this test, it was in the shape of circular crispy chips.

Next, the microstructure of the cross-section (Figure 1(A1–E1)) and flour (Figure 1(A2–E2)) of untreated highland barley grain and highland barley grain after pressing by different thermal treatments (sand-roasting, far-infrared baking, steam explosion, and extrusion) were characterized by SEM. The raw material of highland barley had a smooth cross-section, exhibiting a complete and compact structure. There was no gap in the endosperm. The complete endosperm cell structure could be observed. The structure was wrapped with discoid (type A) and small spherical (type B) starch granules and flocculent crimped protein bodies (Figure 1(A1,A2)), which is consistent with the research results of Yi-peng Bai and Meng-jia Li [7,8]. After different thermal treatments, the internal structure of highland barley changed to varying degrees (Figure 1(B2–E2)). Sand-roasting, far-infrared baking and steam explosion all made highland barley grains have a fluffy structure, and cracks and honeycomb structure appeared in the outer layer and endosperm (Figure 1B–D). Specifically, the highland barley grains treated by steam explosion had a porous honeycomb structure (Figure 1D), the surface area increased, and the starch and protein granules in the endosperm were broken to form a uniform cavity structure (Figure 1(D1)). This result was primarily caused by the evaporation of water in the highland barley grain after the instantaneous release of pressure during steam explosion, indicating better expansibility and cell fragmentation, which was consistent with the research results of Hong Q Y [13]. The volume of highland barley grain after sand-roasting increased significantly, and large cracks and cavities appeared in the outer layer and endosperm (Figure 1B). Compared with untreated highland barley, highland barley retained some complete A-starch granules after sand-roasting, and B-starch granules could hardly be observed. The highland barley fused with the denatured protein to form a disordered network structure, portraying loose and uneven voids (Figure 1(B1)). This was primarily caused by the change of starch granules from crystallization to amorphous following gelatinization [13]. These findings were consistent with the research results of Yi-peng Bai and Altan [8,9]. Starch-based materials were treated at high temperature for a short time, resulting in the “flashing” of liquid water into steam, high internal pressure in the product, and the swelling of starch-based materials in the air. The cross-sectional SEM of highland barley after infrared baking indicated that its volume changed little compared with the untreated highland barley. Obvious swelling occurred near the outer layer, but the endosperm structure was dense without obvious swelling, and only cracks and grooves appeared (Figure 1C). Compared with sand-roasting and steam explosion, relatively complete type A and type B starch granules were retained, and B-starch was obviously exposed due to protein denaturation and shrinkage, forming a relatively orderly arrangement (Figure 1(C1)). One possible reason for this was the high moisture content (40%) of highland barley raw materials pretreated prior to infrared baking in this test. The high moisture content made its matrix soft, and consequently the water vapor reduced the swelling of highland barley grains due to the lack of pressure barrier during heating [28], thus indicating that infrared baking could better retain the tissue structure of highland barley grains. The extrusion treatment changed the grain shape of highland barley (Figure 1E), forming an irregular agglomeration structure with sharp edges and corners and large volume, while the starch granules and protein structure could not be observed at all. The reason for this may lie in the fact that the material was heated and expanded under the action of high temperature and high pressure of the extruder, and the original starch crystal and protein were broken under the action of high shear force, and an adhesive agglomeration structure was formed with other substances after starch gelatinization and protein denaturation [27]. The change in grain structure of highland barley by four thermal treatment methods was bound to affect the physicochemical properties and processing characteristics of its whole grain flour.

### 3.2. Effect of Different Cooking Methods on Crystal Structure of Highland Barley Starch

The information about the relative crystallinity and crystal type was obtained by XRD for the study of starch structure [29]. The results of X-ray diffraction and total relative crystallinity of untreated highland barley flour and heat-treated highland barley flour are demonstrated in Figure 2. As portrayed in Figure 2, the untreated highland barley flour had strong diffraction peaks near 15°, 17°, 18°, 20°, and 23° (2θ). The diffraction peaks near 17° and 18° were connected double peaks; two single peaks appeared at 15° and 23° (2θ); there was a weak diffraction peak at 2θ 20°. The above are the XRD spectrum characteristics of A-type crystalline starch, which are the same as those of seed starch of other cereal crops [7].

In this study, compared with the untreated samples, the XRD peak intensity and crystal structure of whole grain HB flour were significantly improved after thermal treatment. The XRD pattern after infrared baking was like that without treatment, exhibiting a typical A-type crystal structure. The whole grain HB flour treated by sand-roasting, steam explosion, and extrusion presented a typical V-shaped diffraction pattern. This pattern may be the result of complexes formed by the interaction between starch molecules and polar organic molecules during thermal treatment, such as starch-lipid complexes [30], which was similar to the results reported by Wang Lili. The whole grain HB flour treated by steam explosion had strong diffraction peaks near 2θ 5.6°, 15°, and 18°, and weak diffraction peaks near 2θ 20°, thus exhibiting the XRD spectrum characteristics of B-type crystalline starch. The whole grain HB flour treated by sand-roasting had a strong diffraction peak at 2θ 18° and a weak diffraction peak at 20°, which was in line with the XRD spectrum characteristics of A-type crystalline starch. The whole grain HB flour treated by extrusion puffing had only one single peak at 2θ 20°. It was a wide dispersed peak and exhibited the XRD spectrum characteristics of amorphous starch, thus indicating that the spiral structure of highland barley starch changed, the crystal structure had been damaged, and the starch had gelatinized. The crystal structure characteristics of whole grain HB flour with different heat treatments were consistent with the results of the damage degree of highland barley grain structure by heat treatment indicated in SEM (Figure 2). Compared with untreated whole grain HB flour, the crystallinity of whole grain HB flour after infrared baking and instruction puffing was higher, but the crystallinity of whole grain HB flour after sand-roasting and steam explosion was lower. The difference in crystallinity may have been caused by the change in starch granule structure. The reason for this increase in crystallinity may lie in the fact that thermal treatment promoted the formation of new crystals of highland barley starch granules or the recrystallization and perfection of small crystal areas [31], or perhaps the improvement of the enzymatic hydrolysis resistance of starch. The decrease in crystallinity was the result of an increase in the amorphous area of the semi-crystalline layer in starch caused by starch gelatinization under high temperature [32].

### 3.3. Physicochemical Properties

#### 3.3.1. Color

The color characteristics and color parameters of untreated highland barley flour and highland barley flour after different thermal treatments are portrayed in Table 1. The L* value decreased from 79.3 ± 0.015 (untreated highland barley) to 66.88 ± 0.015 (highland barley after steam explosion). There was no significant difference in the L* value of whole grain HB flour after sand-roasting, far-infrared baking, and extrusion (*p* ˃ 0.05), yet higher than that of highland barley flour after steam explosion. This effect may have been caused by the different heating mechanisms of the selected heating methods [7,16]. The decrease in L* value of whole grain HB flour after thermal treatment was primarily caused by the oxidative degradation, browning, and Maillard reaction of anthocyanins and pigments, or other phenolic compounds found in highland barley.

In addition, all treatments in this test increased the a* and b* values of whole grain HB flour from 0.96 ± 0.035 and 11.87 ± 1.205 (untreated highland barley) to 4.46 ± 0.015 and 20.47 ± 0.012 (highland barley after steam explosion), respectively. These changes were attributed to the production of brown substances with different molecular weights at the later stage of the Maillard and caramel reactions, resulting in the increase of a* and b* [7]. To be specific, the a* and b* values of highland barley after steam explosion were the highest. One possible reason for this was that steam explosion caused the greatest damage to the outer layer and endosperm structure of highland barley (Figure 1(D–D2)), and promoted the Maillard and browning reaction of the outer layer of highland barley grain [16]. The total color difference (ΔE) of whole grain HB flour after steam explosion was the greatest (15.51 ± 0.003), being significantly greater than that of the other three thermal treatment methods. The total color difference (ΔE) of whole grain HB flour after sand-roasting, far-infrared baking, and extrusion was not significant (*p* ˃ 0.05), and the total color difference (ΔE) of whole grain HB flour after pressurized puffing was the lowest (9.94 ± 0.128). This was mainly due to the comprehensive influence of different processing methods on different color parameters (e.g., L*, a* and b*) [7]. Different thermal treatments led to the Maillard reaction of highland barley in varying degrees, so that the full whiteness of the cooked whole grain HB flour was reduced, and the color was dark reddish yellow. Additionally, different thermal treatment methods exerted various effects on its color.

#### 3.3.2. Analysis of Thermal Properties and Pasting Properties

The thermal properties of whole grain HB flour with different thermal treatments are portrayed in Table 2. Compared with untreated highland barley flour, the To, Tp, and gelatinization enthalpy (ΔH) of whole grain HB flour after different thermal treatments decreased significantly, and there were significant differences between different thermal treatments (*p* < 0.05). The respective variation ranges were 72.50–34.50 °C, 93.40–64.90 °C, and 7.54.0–2.36 J/g, which was consistent with the results of Zhu’s study [33]. In general, the gelatinization onset temperature is an index by which to measure the perfection of starch microcrystals, and the closer to perfect the microcrystals are, the higher the gelatinization onset temperature will be [34]. Therefore, the results of this test indicated that different thermal treatments destroyed the crystal structure and amorphous area of highland barley starch granules, and the rupture and gelatinization of starch led to the loss in the starch arrangement order, in turn resulting in the gelatinization onset temperature and peak temperature of its flour. The decrease of gelatinization enthalpy (ΔH) indicated that the molecular chain of starch rearranged and formed a new molecular order during thermal treatment [35]. The difference of gelatinization temperature between different thermal treatments may be attributed to the molecular structure (unit chain length and branching degree) and granule structure (crystalline to amorphous ratio) of amylopectin [36]. This result differed from the previously reported result that heat-moisture treatment, superheated steam, microwave, baking, and other treatments improved the To, Tp, and Tc values of sorghum flour and pretreated highland barley flour [34]. The result in the present study may have been caused by the different thermal treatment methods; it could also be related to the influence of thermal treatment on the cooking degree of grain flour. The reported thermal treatment on grain flour was conducted for the purpose of destroying enzymes, thus the flour was not fully cooked, since the purpose of thermal treatment in this study was to prepare cooked highland barley flour. Therefore, the results of this study differed from those previously reported. However, the final gelatinization temperature (Tc) of highland barley starch increased after thermal treatment, which may have been caused by the formation of amylose and lipid-forming complex [33]. The results indicated that the gelatinization temperature and enthalpy (ΔH) of whole grain HB flour could be reduced by changing the crystallinity in thermal treatment. The T0 and Tp of whole grain HB flour after sand-roasting were the lowest, while the Tc and enthalpy (ΔH) of whole grain HB flour after extrusion-puffing were the highest.

Gelatinization characteristics represent the cooking quality and texture behavior of grain flour. Table 2 portrays the gelatinization properties of the control and cooked whole grain HB flour. Compared with the control, the peak viscosity (PV), trough viscosity (TV) and final viscosity (FV) of whole grain HB flour after thermal treatment were significantly reduced (*p* < 0.05). The peak viscosity mainly reflected the ability of binding water or the degree of granule swelling. The higher the peak viscosity was, the stronger the viscosity of the sample would be. The decrease of PV may have been the result of the decrease in starch polymerization degree or starch degradation caused by the rupture of starch chain. The PV value of whole grain HB flour after extrusion puffing was the highest, followed by the samples after sand-roasting and infrared-baking, and the PV values of the sample after steam explosion were the lowest, at 1461, 993, 670, and 425 cp, respectively. The results indicated that the whole grain HB flour after extrusion had stronger water-binding capacity than those treated by other methods, and a strong gel could be formed after water swelling. This may be related to the formation of starch polymer chain, molecular chain rearrangement, and change in crystallinity [37].

The trough viscosity (TV) was related to the ability of starch to resist physical damage during cooling. Compared with the control sample, the TV value of whole grain HB flour treated by sand-roasting was the highest, followed by the samples treated by extrusion and far-infrared baking, and the TV values of the sample treated by steam explosion were the lowest, at 863, 436, 468, and 288 cp, respectively. The results indicated that the whole grain HB flour treated by sand-roasting had strong shear resistance after gelatinization and cooling. FV was related to the rearrangement and reorientation of starch granules, particularly the amylose content. The FV value of the cooked whole grain HB flour was lower than that of the untreated sample, which was attributed to the decrease in the stability of the gel formed by cooling after cooking.

The breakdown viscosity (BD) indicated the thermal stability of whole grain HB flour paste. The greater the BD value was, the worse the thermal stability of whole grain HB flour paste would be. The BD also reflected the fragility of grain flour particles and the stability of paste during heating [20]. After thermal treatment, the BV value of whole grain HB flour was relatively low, thus indicating a high shear-thinning resistance during cooking [38]. The BD values of the samples treated by steam explosion and sand-roasting were the lowest, followed by those of the samples treated by far-infrared baking, and those of the sample treated by extrusion were the highest, at 137, 130, 202, and 1028 cp, respectively. This indicates that the samples treated by steam explosion and sand-roasting had the strongest ability to resist shear thinning during heating.

The setback viscosity (SB) reflected the aging or setback degree of the whole grain HB flour, along with the strength of the gel formed during cooling. The higher the SB value was, the worse the stability of the whole grain HB flour would be after gelatinization and cooling, while the stronger the gel was, the easier the aging would be. The setback value (SB) of the whole grain HB flour treated by steam explosion and extrusion was significantly lower than that of the control sample (*p* < 0.05), at 69 and 146 cp, respectively. The setback value (SB) of the whole grain HB flour treated by sand-roasting and far-infrared baking was significantly higher than that of the control (*p* < 0.05), at 818 and 641 cp, respectively, thus indicating that the whole grain HB flour treated by steam explosion and extrusion did not easily undergo setback and had a strong anti-aging ability. Therefore, steam explosion and extrusion were beneficial to improving the anti-aging ability of highland barley flour and could be used as an anti-aging agent. Therefore, sand-roasting and far-infrared baking are beneficial to improving the gel strength of highland barley flour and can be used as thickening methods.

#### 3.3.3. Functional Characteristics

The functional characteristics such as swelling power, water-holding capacity, oil-holding capacity, and bulk density play important roles in the development of baking and candy products. Swelling power is used to evaluate the swelling behavior of grain starch components in excess water [9]. The swelling powers of untreated whole grain HB flour and heat-treated whole grain HB flour are demonstrated in Table 3. The different thermal treatments improved the swelling power of whole grain HB flour, and a possible reason for such improvement is that the thermal treatment increased the porosity and looseness of the highland barley structure (Figure 1), which is conducive to water absorption by cells in grains [9]. However, various thermal treatment methods bore different effects on the swelling power of whole grain HB flour (Table 3). The swelling power of whole grain HB flour after steam explosion and extrusion was the highest, more than twice that of untreated whole grain HB flour. The swelling power of whole grain HB flour after sand-roasting and far-infrared baking was slightly higher than that of untreated whole grain HB flour, but the difference was not significant (*p* ˃ 0.05). There were large voids and fragments in the tissue of highland barley after extrusion and steam explosion (Figure 1), thereby resulting in greater hydration and increased swelling power [39]. There were few fragments in the tissue of highland barley after far-infrared baking and sand-roasting, and the endosperm structure was relatively complete (Figure 1), thus resulting in less hydration and lower swelling power [37]. In the thermal treatment process, the damaged highland barley starch could absorb more water or unfold the protein so that the previously hidden hydration sites would be exposed, in turn making it easier to interact with water and thus increasing the swelling power [40]. The results indicated that the whole grain HB flour treated by extrusion puffing and explosion puffing was easier to absorb water and swell, and had superior palatability.

Bulk density is an important index to reflect the filling property of flour. The bulk densities of the untreated whole grain HB flour and heat-treated whole grain HB flour are portrayed in Table 3. Compared with untreated whole grain HB flour (the tap density and bulk density were 0.87 mL/g and 0.43 mL/g, respectively), the density of highland barley flour after extrusion (the tap density and bulk density were 0.85 mL/g and 0.70 mL/g, respectively) was significantly higher. For whole grain HB flour in the test, 34% water extrusion was adopted, and the high temperature, high pressure, and high shear force of the extruder made the highland barley flour both rigid and brittle, in turn reducing the fluidity of the flour, and thus improving the bulk density [41]. The tap density of whole grain HB flour after far-infrared baking was 0.55 mL/g and its bulk density was 0.45 mL/g, which had no significant difference from the control (*p* > 0.05). The density of whole grain HB flour after sand-roasting (the tap density and bulk density were 0.45 mL/g and 0.3270 mL/g, respectively) and steam explosion (the tap density and bulk density were 0.48 mL/g and 0.32 mL/g, respectively) was significantly lower than that of the control. This was consistent with the result that thermal treatment reduced the bulk density of highland barley, barley, and wheat reported by Yi-peng Bai, Altan, and Subhamoy Dhua [8,9,16]. The primary reason for this result lies in that the loss of integrity between starch–starch and starch–protein matrices in highland barley due to thermal treatment, or the formation of space in starch endosperm reducing the weight of highland barley grain and the dispersion of flour [17]. Therefore, during the processing of highland barley food, the corresponding cooking method must be selected according to the types and requirements of processed food.

Water-holding capacity and oil-holding capacity have important roles in biscuit and pastry products [16]. The results of water-holding capacity and oil-holding capacity of untreated whole grain HB flour and heat-treated whole grain HB flour are portrayed in Table 1. The various thermal treatments all improved the water- and oil-holding capacities of whole grain HB flour, yet there were differences among the treatments. The water-holding capacities of the whole grain HB flour after extrusion and steam explosion were 3.19 and 3.01 g/g, respectively, both being significantly higher than those after sand-roasting (2.05 g/g) and far-infrared baking (2.06 g/g). The water-holding capacity of whole grain HB flour was related to the number of hydrophilic groups in protein and carbohydrates. The significant increase of WHC after thermal treatment was attributed to new hydrophilic points formed after the gelatinization of starch and denaturation of protein [42]. The gelatinization of the grain flour was affected by processing methods, moisture content and processing conditions, thus resulting in the difference of water-holding capacity between different thermal treatment methods [37]. Similar results were observed in sand-roasted chickpea flour, roasted sorghum flour, and roasted wheat flour [6,16,20].

Compared with untreated whole grain HB flour (1.24 g/g), the oil-holding capacities of whole grain HB flour after steam explosion (3.22 g/g), sand-roasting (1.75 g/g), and far-infrared baking (1.59 g/g) were significantly increased, while extrusion (1.31 g/g) had no significant effect on these values. The change in oil-holding capacity of whole grain HB flour during processing was related to the dissolution and degradation of protein in the flour and the increase or decrease of polar and nonpolar binding sites. Thermal treatment led to protein dissociation in whole grain HB flour, increased the non-polar and polar binding sites, and enhanced its oil-holding capacity [38]. Similar results have been observed in roasted wheat and roasted sweet chestnuts [16,38]. Compared with other thermal treatment methods, extrusion reduced the oil-holding capacity of flour due to the increase of polar sites and the decrease of nonpolar sites in the samples caused by the greater gelatinization effect.

It is thus concluded that the swelling power, water-holding capacity and oil-holding capacity of whole grain HB flour are affected by different thermal treatment methods. Extruded whole grain HB flour is more suitable for manufacturing instant flour products, while sand-roasted and far-infrared baked whole grain HB flour are more suitable for making bakery products such as biscuits and pastries, and steam exploded whole grain HB flour is more versatile and can be used as raw material for instant flour and pastry products.

### 3.4. Storage Characteristics

#### 3.4.1. Moisture Content

The moisture content of grain is related to the weight of dry matter and is a major factor affecting storage stability. Figure 3 portrays the changes in moisture content of the whole grain HB flour before and after thermal treatment and during storage. Compared with the untreated whole grain HB flour, the moisture contents of whole grain HB flour after thermal treatment decreased significantly, all being below 5%. Water is an important medium for heat transfer in the process of thermal treatment. While transferring heat, part of the water vaporizes, and this is the primary reason for the decrease of moisture content in cooked whole grain HB flour [43].

With the extension of storage time, the whole grain HB flour absorbed the water in the environment, and the moisture content gradually increased (Figure 3). After 14 days of storage, the moisture content of whole grain HB flour tended to balance. Grain is a porous colloidal material, with a strong ability to adsorb gas and vapor. In this study, under certain conditions of ambient temperature and humidity, when the water vapor pressure inside and outside the grain or flour was equal, the moisture absorption and moisture dissipation of the sample were in dynamic equilibrium. During the storage period determined by the test, the whole grain HB flour treated by extrusion exhibited a strong ability to adsorb water in the environment, and the ability of the whole grain HB flour after other cooking treatments to adsorb water in the environment was similar, without significant difference (*p* ≤ 0.05). The moisture content of extruded whole grain HB flour reached about 8% on the 14th day. With the extension of storage time, the moisture content of the control gradually decreased. After 56 days of storage, the moisture content tended to balance, and stabilized at about 7%.

#### 3.4.2. Lipase Activity

Improving the storage stability is typically achieved by destroying enzymes or inhibiting enzyme activity. Thermal treatment will effectively reduce lipase activity and enhance grain storage stability. Previous reports have indicated that microwave, hot steam, dry heat, infrared, extrusion, and other thermal treatments exert a good effect on grain lipase inactivation [44]. Figure 4 indicates that the untreated whole grain HB flour had high lipase activity. The lipase activity of whole grain HB flour after different thermal treatments was significantly reduced by 5% (*p* < 0.05), all maintained below 5 mg g^−1^. With the increase of storage time, the lipase activity increased accordingly. The lipase activity of whole grain HB flour treated by extrusion and steam explosion remained high in the first 28 days of storage, then decreased. There was no significant difference among the other treatments, and the lipase activity remained at a low level the entire time. This was consistent with the change trend of moisture content during storage. The water reabsorption of whole grain HB flour reduced the inhibition and improved the lipase activity. The moisture content of whole grain HB flour was kept at a low level of 7% during storage (Figure 3), thereby effectively inhibiting the lipase activity and keeping the lipase activity of cooked whole grain HB flour at a low level during the entire storage period.

#### 3.4.3. Fatty Acid Value

Fatty acid value is a sign to measure the content of free fatty acids in fat, and it can be used as an indicator of the degree of fat hydrolysis during storage. A low fatty acid value signifies that the degree of fat hydrolysis is small, and that the sample is fresh and not prone to rancidity [45]. Figure 5 portrays the changes of free fatty acids during the storage of whole grain HB flour. Compared with the control, except for steam explosion, the fatty acid values of whole grain HB flour after sand-roasting, far-infrared baking, and extrusion were significantly reduced (*p* < 0.05), at 18.8 mg/100 g, 24.33 mg/100 g, and 12.38 mg/100 g, respectively. This result indicated that thermal treatment exerted little effect on the fatty acid value of the whole grain HB flour.

During accelerated storage, the fat in whole grain HB flour was slowly hydrolyzed to produce a certain amount of free fatty acids, resulting in the increase of fatty acid value to varying degrees (Figure 5). With the extension of storage time, the content of free fatty acid in untreated whole grain HB flour increased continuously. The fatty acid value increased significantly after 42 days, reaching 58.89 mg/100 g, and 161.73 mg/100 g after 56 days of accelerated storage, increasing sharply again after 84 days. After steam explosion, extrusion, far-infrared baking, and sand-roasting, the fatty acid value of whole grain HB flour fluctuated slightly in the first 42 days of accelerated storage, which were 102.56 mg/100 g, 23.46 mg/100 g, 53.67 mg/100 g, and 38.38 mg/100 g, respectively, and increased sharply after 84 days. As stipulated by the national safe grain storage standard, grains with fatty acid value of more than 30 mg KOH/100 g should not be stored [45]. Therefore, it is suggested to control the storage period of untreated whole grain HB flour for 10 months at room temperature, with the safe storage period of cooked whole grain HB flour after thermal treatment being 8 months.

#### 3.4.4. Peroxide Value

Peroxide value is an important index by which to reflect the initial oxidative rancidity of fat, mainly focusing on the deterioration of fat during storage [26]. It can be observed from Figure 6 that the peroxide value of whole grain HB flour after thermal treatment was significantly higher than that of the control (*p* > 0.05). The peroxide value of untreated whole grain HB flour was 0.0047 meq/kg. After steam explosion, extrusion, far-infrared baking, and sand-roasting, the peroxide values of the whole grain HB flour were 0.0354 meq/kg, 0.0708 meq/kg, 0.0503 meq/kg, and 0.0800 meq/kg, respectively, indicating that oxidative decomposition of lipid occurred during thermal treatment of highland barley. The reason for this may be that the thermal treatment process induced the decomposition of cell wall, and thus increased the fat-soluble components. Meanwhile, the low water activity broke the bound water on the surface of lipid molecules and increased the lipid oxidation rate [25].

Under the same conditions, the storage time exerted a significant effect on the peroxide value of whole grain HB flour during storage (*p* < 0.05). With the extension of storage time, the fat of whole grain HB flour continued to undergo oxidative rancidity, and consequently, the peroxide value of whole grain HB flour gradually increased during storage (Figure 6), and increased sharply after 56 days of accelerated storage. The peroxide value of whole grain HB flour after extrusion and sand-roasting changed greatly in the accelerated storage process, and the peroxide values after steam explosion and far-infrared baking were relatively stable in the accelerated storage process. This indicates that the whole grain HB flour treated by extrusion and sand-roasting was unstable during storage, while the whole grain HB flour treated by steam explosion and far-infrared baking was relatively stable during storage.

#### 3.4.5. Malondialdehyde Content

With the oxidation of fat, the peroxide produced will degrade and the peroxide value will drop, yet the malondialdehyde value will continue to increase. Consequently, the malondialdehyde content may better reflect the oxidation degree of fat [46]. Figure 7 indicates the change law of malondialdehyde value of whole grain HB flour during storage. Compared with the control, with the increase of storage time, the malondialdehyde value of whole grain HB flour after different thermal treatments fluctuated greatly. In general, it demonstrated an increasing trend, which was consistent with the above change trend of fatty acid value and peroxide value. It was fully demonstrated that the cooked whole grain HB flour after thermal treatment had oxidative rancidity of fat during storage, and the fat oxidation became more apparent with the increase of storage time. Specifically, the malondialdehyde value of whole grain HB flour treated by extrusion and sand-roasting maintained a high level during storage, at 0.0102–0.0270 mg/kg and 0.019–0.2055 mg/kg, respectively, followed by far-infrared baking and steam explosion. This result revealed that the whole grain HB flour treated by extrusion and sand-roasting exhibited a serious degree of fat oxidation and rancidity during storage, and thus was unstable during storage, while the whole grain HB flour treated by steam explosion was relatively stable during storage.

## 4. Conclusions

Sand-roasting, far-infrared baking, steam explosion, and extrusion could effectively alter the appearance and internal starch structure of highland barley, while the XRD of highland barley starch after steam explosion was B-shaped, and the XRD of highland barley starch after extrusion was amorphous. The gelatinization characteristics, swelling power, oil-holding capacity, and water-holding capacity of the flour were altered significantly by thermal treatment. Extruded whole grain HB flour has high water-holding capacity, swelling power, bulk density, the lowest oil-holding capacity, and maximum gelatinization viscosity, thus making it suitable for producing instant flour products. Meanwhile, sand-roasted and far-infrared baked whole grain HB flour has high oil-holding capacity, and is suitable for producing bakery products such as biscuits and pastries. Steam-exploded whole grain highland barley powder has good expansion force, water-holding capacity, and oil-holding capacity, and the bulk density is the smallest, thus it can be used as the processing raw material of high-oiliness baking products. Sand-roasting and far-infrared baking bear relatively little effect on the physicochemical properties of whole grain HB flour. Under the same storage conditions, the untreated whole grain HB flour is stable during storage, and the whole grain HB flour after different thermal treatments is unstable during storage. However, the storage time of the whole grain HB flour treated by steam explosion is the longest, followed by far-infrared baking and sand-roasting, while the storage time of the whole grain HB flour treated by extrusion is the shortest. The physicochemical properties of whole grain HB flour can be obviously improved by steam explosion, and it has better storage stability. In the future, the storage conditions and regulation technologies of cooked whole grain HB flour must be studied to prolong its shelf life, and research in this area is of great significance to the development of whole grain HB food.

## Figures and Tables

**Figure 1 foods-11-02021-f001:**
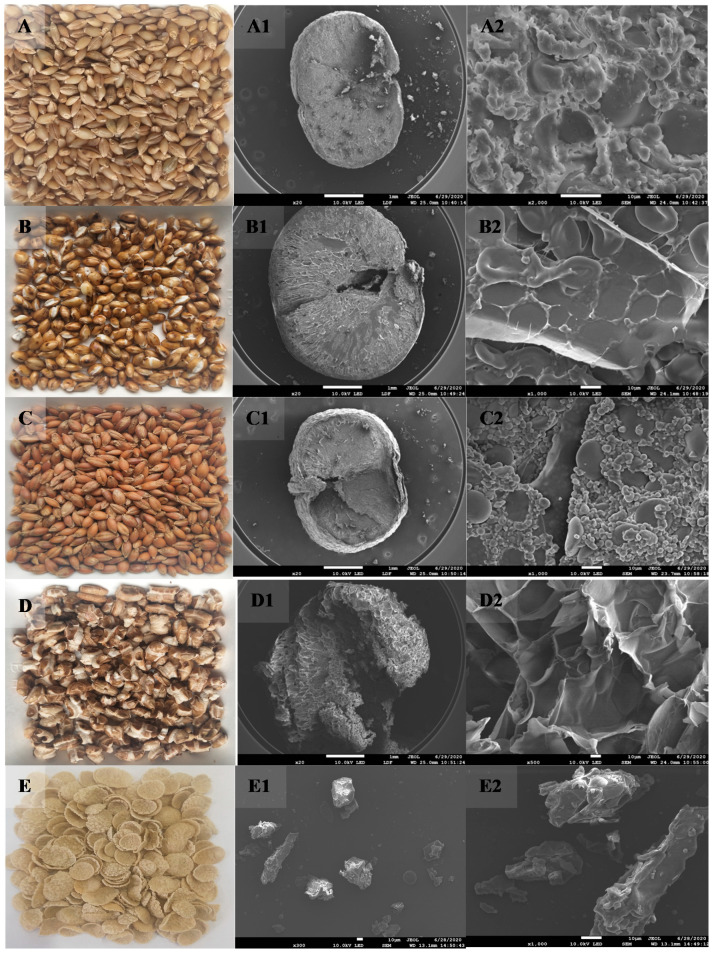
SEM of highland barley after processing by different thermal treatments. (**A**): Untreated (CK); (**B**): Sand-roasting (SR); (**C**): Far-infrared Baking (FB); (**D**): Steam Explosion (SE); (**E**): Extrusion (EX).

**Figure 2 foods-11-02021-f002:**
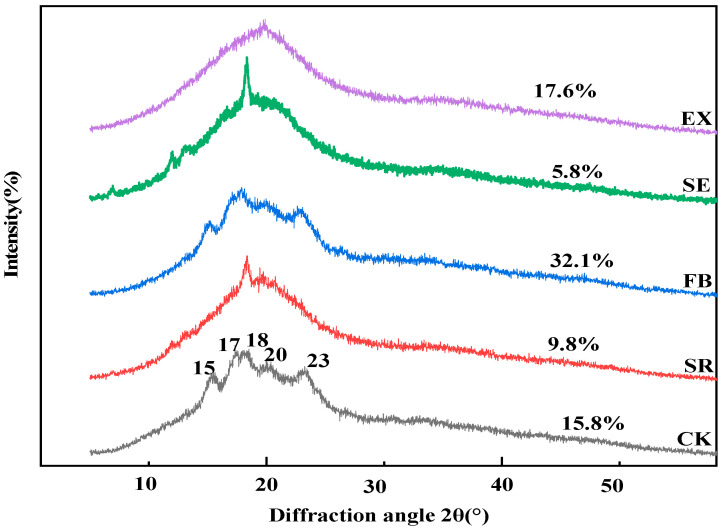
X-ray patterns of whole grain highland barley flour with different thermal treatments. CK: untreatment; SR: Sand-roasting; FB: Far-infraredBaking; SE: Steam Explosion; EX: Extrusion.

**Figure 3 foods-11-02021-f003:**
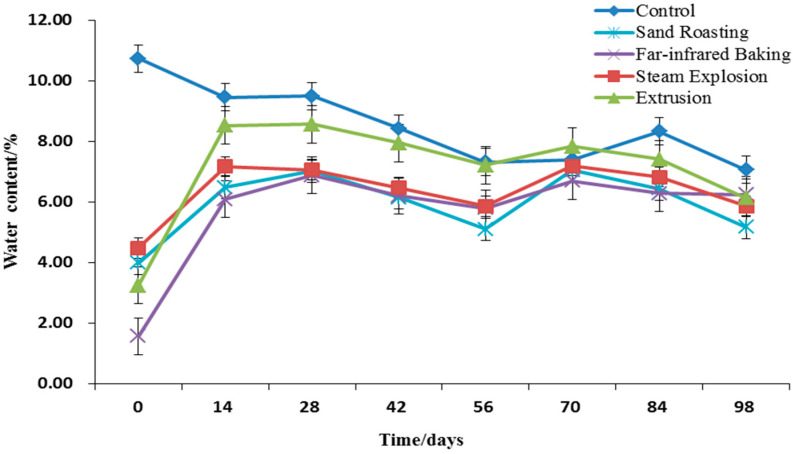
Moisture content of whole grain highland barley flour with different thermal treatments during storage.

**Figure 4 foods-11-02021-f004:**
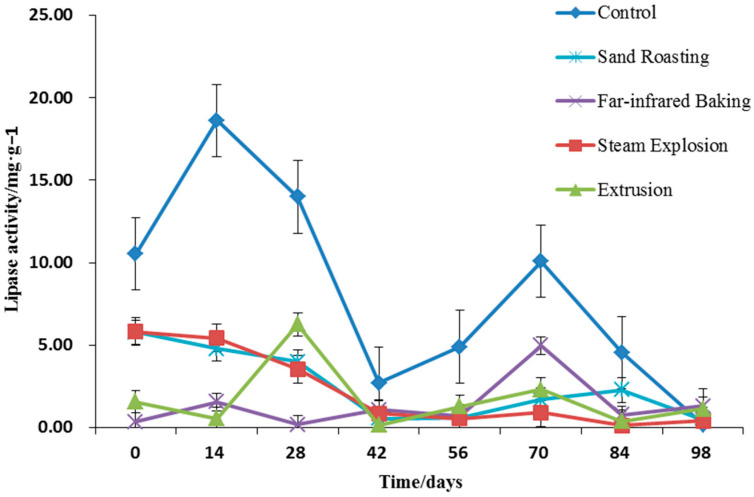
Lipase activity of whole grain highland barley flour with different thermal treatments during storage.

**Figure 5 foods-11-02021-f005:**
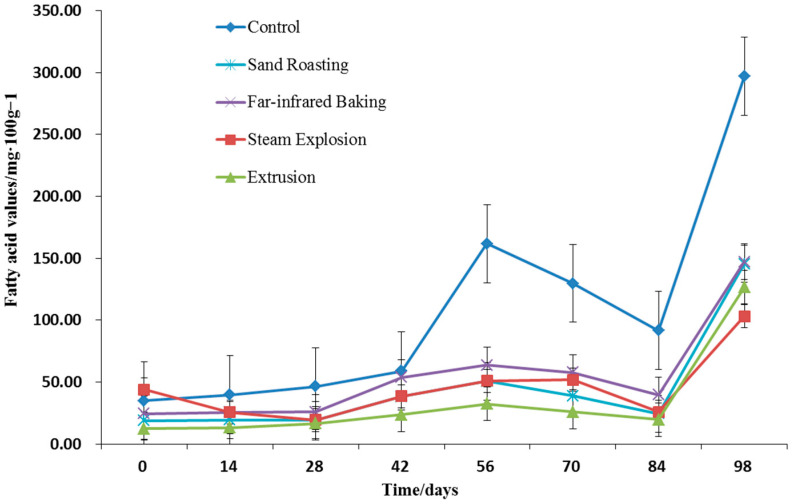
Fatty acid value of whole grain highland barley flour with different thermal treatments during storage.

**Figure 6 foods-11-02021-f006:**
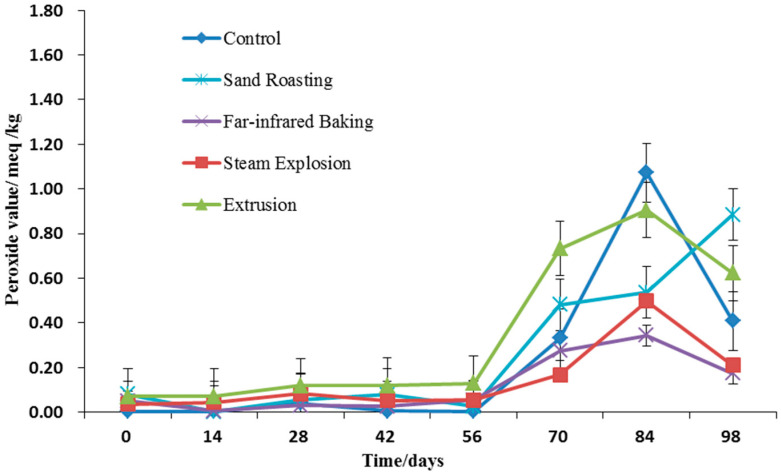
Peroxide value of whole grain highland barley flour with different thermal treatments during storage.

**Figure 7 foods-11-02021-f007:**
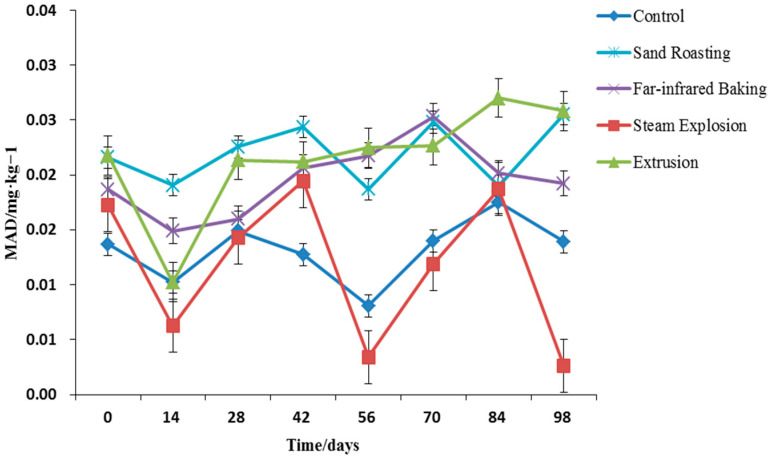
Malondialdehyde content of whole grain highland barley flour with different thermal treatments during storage.

**Table 1 foods-11-02021-t001:** Effect of thermal treatment on colour of whole grain highland barley flour.

Thermal Treatment	Water (%)	L*	a*	b*	△E
Control	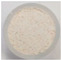	10.73 ± 0.011 a	79.30 ± 0.015 a	0.96 ± 0.035 d	11.87 ± 1.205 c	-
Far-infrared Baking	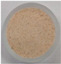	1.57 ± 0.011 d	70.69 ± 0.009 b	1.63 ± 0.025 c	19.17 ± 1.305 a	11.31 ± 0.008 b
Sand-roasting	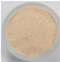	3.99 ± 0.011 c	70.34 ± 0.003 b	3.48 ± 0.013 b	17.13 ± 0.006 b	10.69 ± 0.005 b
Steam Explosion	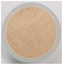	4.48 ± 0.011 b	66.88 ± 0.015 c	4.46 ± 0.015 a	20.47 ± 0.012 a	15.51 ± 0.003 a
Extrusion	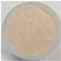	3.25 ± 0.011 c	71.09 ± 0.006 b	1.95 ± 0.009 c	17.39 ± 0.015 b	9.94 ± 0.128 b

Different letters within the same column indicate a significantly different (*p* < 0.05). All values are means of triplicate determinations ± SD.

**Table 2 foods-11-02021-t002:** Thermal and pasting characteristics of whole grain highland barley flour with different thermal treatment.

Thermal Treatment	Thermal Characteristics				Pasting Characteristics				
	T_0_ (°C)	T_p_ (°C)	T_c_ (°C)	△H (J/g)	Peak Viscosity (cP)	Trough Viscosity (cp)	Breakdown Viscosity (cp)	Final Viscosity (cp)	Setback Viscosity (cP)
Control	72.50 ± 0.15 a	93.40 ± 0.31 a	95.90 ± 0.54 a	7.54 ± 1.23 a	2983 ± 60.25 a	2153 ± 20.35 a	830 ± 32.06 b	2501 ± 25.32 a	348 ± 3.53 c
Steam Explosion	52.60 ± 0.10 b	88.60 ± 0.42 b	100.30 ± 0.42 b	2.36 ± 2.31 b	425 ± 9.85 e	288 ± 8.15 d	137 ± 4.89 d	357 ± 9.58 e	69 ± 2.30 e
Extrusion	40.20 ± 0.10 c	73.60 ± 0.11 c	121.70 ± 0.58 c	3.48 ± 0.84 c	1464 ± 50.85 b	436 ± 9.57 c	1028 ± 50.78 a	582 ± 20.42 d	146 ± 3.86 d
Far-infrared Baking	38.00 ± 0.25 c	74.20 ± 0.10 c	110.80 ± 0.60 d	5.39 ± 0.91 d	993 ± 10.25 c	863 ± 10.54 b	130 ± 3.86 d	1681 ± 45.23 b	818 ± 32.06 a
Sand-roasting	34.50 ± 0.21 d	64.90 ± 0.14 d	106.40 ± 0.50 e	7.02 ± 1.25 a	670 ± 22.56 d	468 ± 7.56 c	202 ± 6.53 c	1109 ± 30.41 c	641 ± 10.36 b

Different letters within the same column indicate a significantly different (*p* < 0.05). All values are means of triplicate determinations ± SD.

**Table 3 foods-11-02021-t003:** Effect of thermal treatment on the functional properties of whole grain highland barley flour.

Methods	Swelling Power (%)	Water-holding Capacity (g/g)	Oil-holding Capacity (g/g)	Tap Density (mL/g)	Bulk Density (mL/g)
Control	2.41 ± 0.20 c	1.36 ± 0.12 b	1.24 ± 0.19 b	0.87 ± 0.05 a	0.43 ± 0.03 b
Steam Explosion	6.07 ± 0.10 b	3.01 ± 0.15 a	3.22 ± 0.18 a	0.48 ± 0.05 b	0.32 ± 0.04 c
Extrusion	7.06 ± 0.13 a	3.19 ± 0.16 a	1.31 ± 0.10 b	0.85 ± 0.05 a	0.70 ± 0.03 a
Sand-roasting	2.55 ± 0.17 c	2.05 ± 0.13 b	1.75 ± 0.15 b	0.45 ± 0.03 c	0.32 ± 0.03 c
Far-infrared Baking	2.64 ± 0.19 c	2.06 ± 0.16 b	1.59 ± 0.14 b	0.55 ± 0.03 b	0.45 ± 0.04 b

Different letters within the same column indicate a significantly different (*p* < 0.05). All values are means of triplicate determinations ± SD.

## Data Availability

Data is contained within the article.

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
