# Peer review of "Effect of Thermal Treatment on the Internal Structure, Physicochemical Properties and Storage Stability of Whole Grain Highland Barley Flour"

_foods, 2022, doi:10.3390/foods11142021_

Round 1

Reviewer 1 Report

Figure 2 - Correct "Diffraction" in the axis

Figures 3, 4, 5, 6 and 7 - Indicate the unite for Time (days) in the axis

Reviewer 2 Report

Please rewrite the sentence as it is the first line of Abstract, “to improve the processing performance of whole grain highland barley flour (whole grain HB flour), whole grain HB flour was prepared by sand roasting, far-infrared baking, steam explosion and extrusion”.

Again next line is starting with “the” you can Simply write it as Effects of different thermal treatment methods…..

Authors should add short methodology in the abstract or at least some important parameters.

There is no sequence in whole abstract

Author should add a conclusive line in the end of abstract.

Please mention their values here from previous literature “Highland barley has the characteristics of high protein, high dietary fiber, high vitamins, low fat and low carbohydrates.” In Introduction Line no 42.

Introduction needs serious attention as there are a lot of irrelevant material along with grammatical mistakes.

Add importance and reasoning of this research in the end of introduction session.

Could you mention the reason of soaking Highland barley in sand roasting?

Results are somewhat ok but the length and reasoning should be more relevant.

Conclusion is too lengthy it should be reduced.

Reviewer 3 Report

The document presents the effect of four thermal treatments on the structure, physicochemical properties, and storage characteristics of whole grain HB flour. The research is interesting; however, the authors should reinforce the novelty and originality of this work.

There are some papers that the authors should check:

Bai, Y.-P., et al. (2021). Effect of thermal treatment on the physicochemical, ultrastructural, and nutritional characteristics of whole grain highland barley. Food Chemestry, 346, 128657.

Wang, H., et al. (2022). Different thermal treatments of highland barley kernel affect its flour physicochemical properties by structural modification of starch and protein. Food Chemistry, 387, 132835.

Also, below there are some specific comments that the authors should attend:

I recommend that the authors should include internal structure in the title.

In the abstract section, the authors should define To and Tp abbreviations in the text, also, they should delete P<0.05.

I recommend that the authors should change the keywords (highland barley, thermal treatment, physicochemical properties, and storage stability) because they are repeated in the title.

In the introduction section, the authors omitted a reference (line 82.). In addition, they should check the grammar in this section.

 In lines 113-115, the authors mention that “research regarding the structural changes, physicochemical properties and storage characteristics of cooked highland barley flour after thermal treatment has not yet been reported”, however, they should reconsider the statement about the novelty and contribution of this research to the state of art, since there is a work that has already reported “Effect of thermal treatment on the physicochemical, ultrastructural and nutritional characteristics of whole grain highland barley”.

In lines 127-129, the authors should include the proximal composition of the raw materials. In addition, they should indicate the way to store the samples before their analysis.

In sample preparation (line 130), the authors have references to pre-test the conditions of each thermal method. The samples were preconditioned to specific moisture content, however, this parameter was different in each thermal treatment.     

In lines 319-320, the authors mentioned that “the starch and protein granules in the endosperm were broken…”, can the authors demonstrate this statement?  

In lines 409-411, the authors indicate the thermal effect on color parameters and they mention this effect can be caused by oxidative degradation, browning, and Maillard reaction. Should the authors measure the anthocyanin and phenolic contents in their samples?  

The authors omitted the reference in line 464.

Figures 3, 4, 5, 6, and 7 should include letters to indicate a significantly different.

In general, the document lacks innovations and appears to be a repetition of other past research with a mere change of thermal method.

 Author Response

Reviewer 4 Report

Dear Authors,

Although a lot of experimental work was performed providing meaningful results, the authors need to correct English and more carefully spell-check the manuscript text. The other comments are below.

Regards

Effect of Thermal Treatment on the Physicochemical Properties and Storage Stability of Whole Grain Highland Barley Flour

The aim of this research was to investigate the effects and differences of four thermal treatment technologies: infrared baking, sand roasting, explosion puffing and extrusion puffing, on the structure, physicochemical properties and storage characteristics of whole grain HB flour, so as to provide a theoretical basis for the quality improvement of whole grain HB flour and its application in food development.

Abstract, title and references         

Abstract is clearly written with a good command of English and a clear representation of the aim of the paper. However, containing 324 words, its length goes beyond the propositions of the journal Foods (200 max), so it definitely needs to be shortened, highlighting the most important aspects of this research.

 The title of the paper adequately reflects the subject under investigation in the proposed study.

 References are numbered in order of appearance in the text, as demanded by formatting rules of the journal. Although there is no limitation in the number of references, a reference list of 54 citations is completely sufficient, or even too long to cover the topic proposed. The authors might try to omit references older that 10 years.

Introduction 

The authors clearly represented the importance of the issue described. At the end of the introduction section a clear and concise aim of this study is given.

Line 82: Please, add the reference.

Materials and Methods       

The authors adequately described the sample collection, sample preparation, the methods to analyze whole powder characteristics, functional properties of whole powder, storage stability and statistical methods used.

Lines 224-229: Please, rewrite this section and try to omit using imerative form in the manuscript text.

Results and Discussion        

The results and discussion sections have been shown jointly, as the journal guidelines are not specific and exclusive regarding this issue. The meaning, relevance and importance of the obtained results are explained in the same section. They were adequately compared with previous studies.

Figure 2. Please, correct the word Diffraction.

Line 278: Please, delete the word subsection from the subsection title.

Conclusions  

Conclusions are supported by the results obtained.

Round 2

Reviewer 2 Report

I am agree with these changes

Reviewer 3 Report

I accept the document to publish it in Foods.